



# Using satellite data to identify the methane emission controls of South Sudan's wetlands

Sudhanshu Pandey[1], Sander Houweling[1,2], Alba Lorente[1], Tobias Borsdorff[1], Maria Tsivlidou[3], A. Anthony Bloom[4], Benjamin Poulter[5], Zhen Zhang[6], Ilse Aben[1]

[1]SRON Netherlands Institute for Space Research, Utrecht, the Netherlands
[2]Department of Earth Sciences, Vrije Universiteit Amsterdam, Amsterdam, the Netherlands
[3] Laboratoire d'Aérologie, Université de Toulouse, CNRS, UPS, IRD, France
[4]Jet Propulsion Laboratory, California Institute of Technology, Pasadena, CA, 91109, USA
[5]NASA Goddard Space Flight Center, Greenbelt, MD, 20771, USA
[6]Department of Geographical Sciences, University of Maryland, College Park, MD, 20740, USA

*Correspondence:* Sudhanshu Pandey (s.pandey@sron.nl)

**Abstract.** The TROPOspheric Monitoring Instrument (TROPOMI) provides observations of atmospheric methane ($CH_4$) at an unprecedented combination of high spatial resolution and daily global coverage. Hu et al. (2018) reported unexpectedly large methane enhancements over South Sudan in these observations. Here we assess methane emissions from the wetlands of South Sudan using two years (December 2017–November 2019) of TROPOMI total column methane observations. We estimate annual wetland emissions of $7.2 \pm 3.2$ Tg yr$^{-1}$, which agrees with the multiyear GOSAT inversions of Lunt et al. (2019) but is an order of magnitude larger than estimates from wetland process models. This disagreement may be explained by the up to 4 times underestimation of inundation extent by the hydrological schemes used in those models. We investigate the seasonal cycle of the emissions and find the lowest emissions during the June-August season when the process models show the largest emissions. Using satellite altimetry-based river water height measurements, we infer that this seasonal mismatch is likely due to a seasonal mismatch in inundation extent. In models, inundation extent is controlled by regional precipitation, scaled to static wetland extent maps, whereas the actual inundation extent is driven by water inflow from rivers like the White Nile and the Sobat. TROPOMI emission estimates show better agreement, in terms of both seasonal cycle and annual mean, with model estimates that use a stronger temperature dependence. This suggests that temperature might be the best explanatory control for the emissions from wetlands in South Sudan. Our findings demonstrate the use of satellite instruments for quantifying emissions from inaccessible and uncertain tropical wetlands, providing clues for improvement of process models, and thereby improving our understanding of the currently uncertain contribution of wetlands to the global methane budget.



## 1 Introduction

Reducing anthropogenic methane emissions has been recognized as an important requirement for achieving the 2015 Paris
Agreement target of limiting global temperature rise below 2° C relative to pre-industrial times (Ganesan et al., 2019).
However, large uncertainties remain in the atmospheric budget of methane, calling for an improved understanding of its
emissions from both anthropogenic and natural sources (Saunois et al., 2016). Wetlands are ecosystems with seasonally or
permanently inundated or saturated soils, including peatlands (bogs and fens), mineral wetlands (swamps and marshes), and
seasonal or permanent floodplains, where methanogens produce methane in the anaerobic decomposition of organic matter.
Emissions from natural wetlands are the largest and the most uncertain emission category of methane (Kirschke et al., 2013).
Saunois et al. (2016) provide global methane emission estimates for all source categories combined of 540–568 Tg yr$^{-1}$ using
top-down approaches and 596–884 Tg yr$^{-1}$ using bottom-up approaches for the period 2003–2012. They attribute the mismatch
between the two approaches mainly to uncertainties in emissions from natural wetlands, inland waters and geological sources.
They report total emissions of 127–202 Tg yr$^{-1}$ and 153–227 Tg yr$^{-1}$ from wetlands using top-down and bottom-up approaches,
respectively, which accounts for 30 % of global emissions.

In addition to being an important source of uncertainty in methane budget, wetlands emissions can have significant climate
feedback due to their sensitivity to changes in precipitation and temperature (Arneth et al., 2010; Zhang et al., 2018; Zhu et
al., 2017). By analyzing surface and satellite measurements of methane, Pandey et al. (2017) reported enhanced methane
emissions from tropical wetlands due to precipitation and temperature anomalies associated with the La Niña of 2010.
Furthermore, according to Zhang et al. (2017), the feedback of methane should be accounted for in climate mitigation policies
as they find that the global wetland emissions will increase by 50 to 170 Tg yr$^{-1}$ at the end of the 21st century because of the
temperature-driven increase in wetland emissions under the different Representative Concentration Pathways (RCP) adopted
by the IPCC. Their results indicate that the increase in wetland emissions maybe 38–56 % larger than the projected
anthropogenic emission increase by the end of the 21$^{st}$ century.

To improve wetland emission estimates is very challenging due to many reasons. Wetlands are spread over large, inaccessible
regions around the world. Upscaling a few localized measurements of wetland emissions is often futile as these emissions have
large temporal and spatial variability, and the parameters controlling them are very uncertain. The emissions are also difficult
to monitor on the ground due to logistical limitations. This makes satellite observations a promising and crucial source of
information to advance our understanding of the role of wetland methane emissions in the carbon cycle.

Hu et al. (2018) observed large methane enhancements over South Sudan in TROPOMI data collected during the first two
months of the commissioning phase of the satellite, November and December 2017. Frankenberg et al. (2011) also observed
an enhancement over the region in a seven-year average (2003–2010) of SCanning Imaging Absorption spectroMeter for
Atmospheric CHartographY (SCIAMACHY) observations. These studies indicated that the enhancements are likely caused





by large emissions from wetlands in the region. Recently, Lunt et al. (2019) used methane observations from the Japanese Greenhouse gases Observing Satellite (GOSAT) in inverse modelling to infer emissions from tropical Africa during 2010–2016. They found that emissions from South Sudan were more than 3 times larger than the ensemble mean estimates from the Wetcharts process model (Bloom et al., 2017). They also found that emissions from the Sudd wetlands in the region increased

rapidly from 2.4–4.2 Tg yr$^{-1}$ in 2010–2011 to 5.2–6.9 Tg yr$^{-1}$ in 2016, likely, because of an inundation extent (IE) expansion due to an increase in water inflow from the White Nile river.

This study aims to infer the scale of the wetland methane emissions from South Sudan from TROPOMI observations using a simplified emission quantification method and investigate its relationship with the results of wetland process models and the seasonally varying climatological conditions. This study is structured as follows. Section 2 describes the method and data used

including the TROPOMI data, wetland models and IE data, and the emission quantification method. Section 3 presents our results and discussion including emission estimates from TROPOMI and their comparison with the process models, and an analysis of the differences between models and TROPOMI emission estimates using IE and meteorological data. Our conclusions are given in Section 4.

## 2 Data and method

### 2.1 TROPOMI methane data

TROPOMI is the single instrument onboard the Copernicus Sentinel-5 Precursor (S-5P) satellite, launched on 13 October 2017 in a sun-synchronous orbit at 824 km altitude (Veefkind et al., 2012). It is a push-broom imaging spectrometer, recording spectra along a 2600 km swath while orbiting the Earth every 100 min, resulting in daily global coverage. Total column methane (XCH$_4$) is retrieved with near-uniform sensitivity in the troposphere from its absorption band around 2.3 $\mu$m using

earthshine radiance measurements from the Short Wave Infrared (SWIR) channel of TROPOMI (Hu et al., 2016; 2018). TROPOMI XCH$_4$ has a ground pixel size of $7 \times 7$ km$^2$ ($7 \times 5.6$ km$^2$ since August 2019) at nadir with larger ground pixels towards the edges of its swath.

In this study, we use the operational two-band retrieval product of TROPOMI (Hasekamp et al., 2019). It uses 0.76 $\mu$m O$_2$A

and 2.3 $\mu$m CH$_4$ bands in the Near Infrared (NIR) and SWIR spectra. XCH$_4$ is retrieved using the full-physics RemoTeC algorithm, which accounts for light path perturbations due to scattering by aerosol and cirrus cloud particles in the atmosphere (Butz et al., 2012; Hu et al., 2016). We only use high-quality XCH$_4$ measurements retrieved under favourable cloud-free conditions. Also, XCH$_4$ is filtered ("$q$a"=1) for solar zenith angle ($< 70°$), viewing zenith angle ($< 60°$), smooth topography (1-standard deviation surface elevation variability $< 80$ m within a 5 km radius) and low aerosol load (aerosol optical thickness

$< 0.3$ in SWIR band). Note that Hu et al. (2018) used two months of XCH$_4$ data from the "scientific" retrieval product of SRON Netherlands Institute for Space Research. Those measurements had a relatively sparse temporal coverage over South Sudan because they were performed during the commissioning phase of TROPOMI when algorithm tests and calibrations were



ongoing. The operational product used here provides a more temporally homogenous coverage and a surface albedo-dependent bias correction (Hasekamp et al., 2019).

## 2.2 Process model data

We compare TROPOMI emission estimates with two wetlands process models: Wetcharts (Bloom et al., 2017) and LPJ-wsl (Zhang et al., 2016). These models calculate monthly methane emissions on a global grid of $0.5° \times 0.5°$ resolution by simulating the microbial production and oxidation processes in the soil using temperature, IE and heterotrophic respiration data. Wetcharts calculates wetland emissions using in total four IE parameterizations, nine terrestrial biosphere models of heterotrophic respiration and three $CH_4$:C temperature parameterizations (q10). Wetcharts version 1.0 provides two ensembles: (1) an ensemble with 324 emission estimates for 2009-2010, called the "Full Ensemble" and (2) an 18-member extended-in-time ensemble for 2001-2015, called the Wetcharts "Extended Ensemble". In the Wetcharts Full Ensemble, the set of four IE estimates are calculated based on two maximum wetland area estimates, multiplied with two monthly varying scaling factors. The wetlands area estimates are taken from (1) the Global Lakes and Wetlands Database (GLWD; Lehner and Döll, 2004), and (2) the sum of all GLOBCOVER wetland and freshwater land types (Bontemps et al., 2011). The scaling factors are calculated from (1) precipitation data from ERA-Interim meteorological data and (2) IE data from the Surface WAter Microwave Product Series (SWAMPS) multi-satellite surface water product (Schroeder et al., 2015). The 18-member Wetcharts Extended Ensemble provides emission estimates for only the two IE estimates that are based on ERA-Interim and only one terrestrial biosphere model CARDAMOM (Bloom et al., 2016).

LPJ-wsl methane model is based on the process-based dynamic global vegetation model Lund Postdam Jena (LPJ). It uses soil temperature, soil moisture-dependent fraction of heterotrophic respiration ($R_h$), and IE to calculate wetlands methane emissions. The IE of LPJ-wsl is calculated by the TOPography-based hydrological model (TOPMODEL) driven by meteorology from ERA-Interim. TOPMODEL simulates hydrologic fluxes of water, including lateral transport, such as infiltration-excess overland flow, infiltration, exfiltration, subsurface flow, evapotranspiration, and channel routing through a watershed.

## 2.3 Inundation extent data

Earlier studies have indicated that the water availability is particularly important in the tropics (temperature is less limiting here in contrast to high latitudes), and hence, IE is one of the main sources of uncertainty for tropical wetlands (Bloom et al., 2010; Ringeval et al., 2010). We analyze the IE data used in process models: TOPMODEL (used in LPJ-wsl); GLWD and GLOBCOVER with ERA-Interim (used in the Wetcharts Extended Ensemble). We compare these IE estimates against the remote sensing-based high-resolution IE data from Gumbricht et al. (2017), which maps wetlands and peatlands at 231 meters spatial resolution by combining three biophysical indices related to wetland and peat formation: (1) long-term water supply exceeding atmospheric water demand, (2) annually or seasonally waterlogged soils, and (3) geomorphological position where





water is supplied and retained. They use 2011 MODIS data to map the duration of wet and inundated soil conditions and
Shuttle Radar Topography Mission (SRTM) for topography. In addition, we use satellite altimetry-based water height
measurements from the Hydroweb database (Crétaux et al., 2011; Da Silva et al., 2010). The water height anomalies of the
White Nile and Sobat rivers are used as a proxy for IE variations in the Sudd and Macher wetlands, respectively. Fig. 1b shows
the location of the river height measurement sites. We also analyze temperature and precipitation data from the European

Centre for Medium-Range Weather Forecasts' ERA5 meteorological reanalysis (Hersbach and Dee, 2016).

## 2.4 Emission Quantification method

The wetland distribution from Gumbricht et al. (2017) is shown in Fig. 1 for the region in South Sudan where a large TROPOMI
$XCH_4$ enhancement can be observed. This region, which includes Sudd, Machar and other smaller wetlands, is hereafter
referred to as the South Sudan wetland region (SSWR). We apply the mass balance method of Buchwitz et al. (2017) to

seasonally averaged TROPOMI $XCH_4$ to calculate emissions from December 2017 to November 2019. The emission $Q$ (Tg
$yr^{-1}$) from the SSWR box in Fig. 1a for a given period is calculated using the following equation:

$$Q = \Delta XCH_4 \times M \times M_{exp} \times L \times V \times C \quad (1)$$

Where, $\Delta XCH_4$ is the "source $XCH_4$ enhancement", i.e., the mean $XCH_4$ difference between the source and the surrounding
background. $C$ is a dimensionless factor of 2.0 derived by Buchwitz et al. (2017) based on the concentration difference of air
parcels before and after entering a source area. $M$ (5.345 Tg $CH_4$ $km^{-2}$ $ppb^{-1}$) is the atmospheric total column mixing ratio-to-
mass conversion factor for a surface pressure of 1013.25 hPa, which is the standard atmospheric pressure. $M_{exp}$ is a
dimensionless factor used to correct for the changes in column air mass with surface elevation, calculated as the ratio of surface

pressure in the source and standard atmospheric pressure (1013.25 hPa). $L$ is the "effective size" of the source region (632
km), calculated as the square root of its area ($4.0 \times 10^5$ $km^2$). $V$ (km $yr^{-1}$) is the ventilation wind speed derived from the ERA5
meteorological reanalysis vertical wind speed profile. Surface elevation variations change the contribution of tropospheric to
the total atmospheric column, which influences $XCH_4$. TROPOMI $XCH_4$ maps are corrected for this effect by adding the
correction factor 7 ppb $km^{-1}$ from Buchwitz et al. (2017), using GMTED2010 elevation data shown in Fig. A1 (Danielson et

al., 2010).

Figure 2 shows the monthly average ERA5 wind speed at 10:00 UTC (TROPOMI overpass time) in the SSWR as a function
of pressure within the local boundary layer during select months. To calculate $V$, the pressure-weighted average of these
boundary layer wind speed is calculated over SSWR using monthly average ERA5 boundary layer height data. We use average

boundary layer winds instead of 10-meter winds because it was found to better represent the ventilation wind speed in the





source region (see Varon et al., 2018). For SSWR, the 10-meter wind speed is on average 35 % lower than the boundary layer wind speed, consistent with the diminishing influence of the surface friction with height.

The uncertainty of $Q$ is calculated as sum-in-quadrature of uncertainties associated with $\Delta XCH_4$ and $V$. The $\Delta XCH_4$ uncertainty

is estimated as sum-in-quadrature of (1) 1-standard deviation of $\Delta XCH_4$ estimates calculated by sequentially increasing the size of the background box from 1° to 10° longitude and latitude in 1° interval, and (2) the XCH4 uncertainty of a single 0.1° × 0.1° grid cell in the average map (= 22 ppb), taken as 1-standard deviation XCH4 of all the grid cells in Fig. 1a. Note that this approach overestimates the XCH4 uncertainty of the grid cells as XCH4 variations within the grid are also caused by emissions and surface elevation variations in addition to measurement errors. The uncertainty of $V$ is estimated from the

variation in wind speed during 4 consecutive hours (09:00 UTC, 10:00 UTC, 11:00 UTC, 12:00 UTC) centred around the TROPOMI overpass time. Note that we use the mass balance method equation from Buchwitz et al. (2017), but not their empirical equation to estimate the uncertainty of $Q$. They derive that equation using a fixed wind speed of 1.1 m/s globally, which would give a larger uncertainty in comparison to our approach of using location and time-specific wind information: ERA5 average $V$ is $2.5 \pm 0.42$ m s$^{-1}$ in SSWR during 2018-2019.

**3 Results**

**3.1 XCH4 enhancements**

We first assess the XCH4 enhancements over South Sudan in the two-year average map of TROPOMI XCH4 shown in Fig. 1a in relation to the SSWR wetland distribution in Fig. 1b. Similar to previous remote sensing studies (Frankenberg et al., 2011; Lunt et al., 2019; Hu et al., 2018), we observe a large XCH4 enhancement over the Sudd wetlands. In addition, the TROPOMI

data also resolve another distinct enhancement over the Machar and Lotilla wetlands in eastern South Sudan, indicating large emissions from these wetlands too. The second enhancement was also observed by Hu et al. (2018) using two months of TROPOMI XCH4.

The Sudd wetlands are flooded by the main White Nile tributary originating from Lake Victoria, whereas the wetlands in

Southeast Sudan are along smaller rivers like the Kangen and Sobat, originating from the Ethiopian mountains. Lunt et al. (2019) attributed their GOSAT inversion emission estimates only to Sudd and evaluated the emissions using auxiliary data for Sudd. However, as the wetlands in the east are flooded by a different set of rivers and have a substantial contribution to the overall XCH4 enhancement, they also need to be considered when studying the mechanisms driving the large emissions in this region.


The XCH4 enhancement for SSWR in the two-year average is $18.8 \pm 2.8$ ppb, which is more than 3 times the enhancement over the Permian basin in the USA as reported by Zhang et al. (2020). It is very unlikely that the SSWR enhancement is an





artefact of the known aerosol or surface albedo biases in the TROPOMI XCH₄ data. We elaborate further on this in Appendix Sect. A1. Figure 3 shows seasonally average XCH₄ maps over SSWR, and Table 1 quantifies the seasonal XCH₄ enhancement

and areal coverage of the TROPOMI data. TROPOMI has good coverage in SSWR, ranging from 40 % in JJA to > 90 % DJF. It is higher than 70 % in all seasons except JJA, likely due to persistent cloud cover during the wet season. The lowest enhancements are observed in JJA in both 2018 (7.3 ± 2.4 ppb) and 2019 (-1.5 ± 2.4 ppb). It is unlikely that these low enhancements are artefacts of the low coverage as there is still sufficient TROPOMI data over the areas with large emissions in SSWR, the Sudd and Machar wetlands. SON-2019 has the largest enhancement of 26.3 ± 2.2 ppb, while  DJF-2018 and

MAM-2018 also have large enhancements of about 22 ppb.

**3.2 Emissions quantification**

We use the mass balance method of Buchwitz et al. (2017) to estimate emissions from SSWR for each season (see Table 1). The lowest emissions are in JJA, corresponding to the lowest XCH₄ enhancements. Emissions during other seasons are close to 9 Tg yr⁻¹ except for DJF-2018. Direct application of the mass balance method on the two-year average XCH₄ map shown in

Fig. 1a yields an annual emission of 10 ± 1.7 Tg yr⁻¹. However, this is likely an overestimate as the two-year average temporally under samples the low emissions of JJA seasons due to low coverage during these seasons. Therefore, to ensure uniform temporal sampling of all seasons, we calculate annual SSWR emissions by averaging the seasonal emission estimates, resulting in 8.0 ± 3.2 Tg yr⁻¹. Moreover, this approach is likely less sensitive to error due to mean-of-products vs product-of-means effect. A caveat of the mass balance method is that it ignores two factors: (1) the influence of emissions in the background

region, and (2) the contribution of emissions in the source region to the background average XCH₄. Both factors increase the background XCH₄ and ignoring them results in an underestimation of the emission estimate. However, this underestimation is large when the ratio of the area of the background region and the source region is small, and as we apply the method using a large background, we do not expect a significant impact on our emission estimates.

Lunt et al. (2019) report methane emissions for all sources (including wetlands, biomass burning, anthropogenic, wild animals) from the Sudd wetlands using multiyear GOSAT inversions. Their emission estimate of 5.2–6.9 Tg yr⁻¹ for 2016 is within the uncertainty bounds of our SSWR total emission estimate of 8.0 ± 3.2 Tg yr⁻¹ for 2018-2019. Note that some difference in the emission estimates can be explained by the difference in the definition of the source region between the two studies as their region extends more north and less east than ours. TROPOMI shows a large XCH₄ enhancement over Lottila and Machar

wetlands in the east SSWR, indicating large emissions from these wetlands. As the source region in Lunt et al. (2019) only partially covers these wetlands, our emission estimates are expected to be higher.

To calculate wetlands emissions from SSWR, we account for other methane emissions in the region using bottom-up data. According to the EDGAR (version 4.3.2, Janssens-Maenhout et al., 2017) inventory for 2012, the total anthropogenic

emissions from SSWR were 0.43 Tg yr⁻¹ with enteric fermentation (0.36 Tg yr⁻¹ ) being the largest anthropogenic category.



The region has small emissions from wastewater management (0.03 Tg yr$^{-1}$), energy for buildings (0.01 Tg yr$^{-1}$) and manure management (0.01 Tg yr$^{-1}$). EDGAR does not report any significant emissions from fossil fuel exploitation sector in the region. Recently, Scarpelli et al (2020) presented a new inventory for the oil and gas sector in which UNFCCC reported national emissions are spatially allocated to the fossil fuel infrastructure. They report 0.05 Tg yr$^{-1}$ emissions from SSWR in 2016.

Biomass burning is the largest natural methane source after wetlands in SSWR with average emissions of 0.20 Tg yr$^{-1}$ in 2018-2019 according to GFED4s (0.23 Tg yr$^{-1}$ in 2018, 0.16 Tg yr$^{-1}$ in 2019, Van der Werf et al., 2017). Another significant natural source is the emission from termites (0.16 Tg yr$^{-1}$, Sanderson, 1996). We subtract the total of these non-wetlands emissions to calculate wetland emissions of 7.2 ± 3.2 Tg yr$^{-1}$ from SSWR in 2018-2019. This estimate is an order of magnitude larger than the 0.5 Tg yr$^{-1}$ wetlands emissions from the prominent Pantanal wetlands of South America in 2010–2018 which are estimated

using GOSAT inversions by Tunnicliffe et al. (2020).

Our SSWR wetlands emission estimate of 7 ± 3.2 Tg yr$^{-1}$ can be an overestimate if the emissions from the above-mentioned non-wetlands sectors are underestimated in the inventories. However, this is unlikely as it would require a very large underestimation in the inventories for the two years studied here. For example, for the oil and gas sector, the annual emissions

(0.05 Tg yr$^{-1}$) will need to be underestimated by two orders of magnitude to have a significant error impact on the wetland emission estimates. Moreover, the strong seasonality shown by the TROPOMI emission estimates is not expected in oil and gas emissions. The SSWR biomass burning emissions are larger in comparison to the other sectors, but a large underestimation in annual emissions by GFED is unlikely as it uses remote sensing-based fire activity and vegetation productivity data.

### 3.3 Comparison to wetland process models

### 3.3.1 Annual means

SSWR integrated methane emission estimates from the process models are an order of magnitude lower than those from TROPOMI (Table 2). For example, the multiannual mean emission from LPJ-wsl for 1980-2016 is 1.1 Tg yr$^{-1}$ (ranging from 0.5 Tg yr$^{-1}$ in 1990 to 1.5 Tg yr$^{-1}$ in 1998). The multi-annual mean individual ensemble estimates from Wetcharts Extended Ensemble range from 0.4 Tg yr$^{-1}$ (uses GLWD) to 1.8 Tg yr$^{-1}$ (uses GLOBCOVER). Its smallest and largest annual emission

estimates from these ensemble members are 0.29 Tg yr$^{-1}$ in 2009 and 2.21 Tg yr$^{-1}$ in 2013. The Wetcharts Full Ensemble, with 324 estimates for 2009-2010, has a mean of 0.9 Tg yr$^{-1}$, ranging from 0.07 to 3.7 Tg yr$^{-1}$.

Table 2 also presents the maximum IE extent in the IE data used by the process models, which range from 25,000 to 69,000 km$^2$ across the models. These IE are up to 4 times lower than the observation-based IE estimates of 99,000 km$^2$ by Gumbricht

et al. (2017). Huges & Huges (1992) give the permanent wetland area of the different wetlands in SSWR (Table A1). The sum of these areas is 36,000 km$^2$, significantly larger than the permanent IE area (i.e., minimum mean monthly IE) used in the models (Wetcharts Extended Ensemble: 1,000 km$^2$; LPJ-wsl: 14,000 km$^2$; SWAMPS: 16,000 km$^2$). Rebelo et al. (2011) used





remote sensing data to characterize IE of the Sudd wetlands over a 12 months period, yielding a total wetlands area of 50,000 km² (41,000 km² of seasonally inundated and 9,000 km² of permanent inundated). According to Huges & Huges (1992), other

wetlands in the SSWR have a total permanent wetlands area of >20,500 km², meaning that Sudd accounts for only about a third of the SSWR's total wetland area. As other wetlands in SSWR are also along rivers like Sobat, their IE likely has a large seasonality, and assuming that the relative seasonal amplitude of IE of these other wetlands is similar to that of Sudd would give a total (seasonal + permanent) flooded area of 134,000 km². Adding the Sudd IE yields a total SSWR IE of 164,000 km², which is larger than the total IE estimate of 99,000 km² from Gumbricht et al. (2017). Overall, we find substantial evidence of

underestimations of SSWR IE in the process models, which may explain their emission underestimations as they assume that IE is a strong control of the emissions.

We now look at variations in annual mean IE to find a possible cause of high emissions in 2018-2019. Lunt et al. (2019) attribute the emission increase in South Sudan between 2010 and 2016 to an IE increase in the Sudd owing to an increased

water inflow from the White Nile river found in satellite altimetry-based river water height measurements. To investigate this for the period 2018-2019, we look at trends in water height (see Fig. 4) of Lake Victoria, and White Nile and Sobat rivers. Similar to Lunt et al. (2019), we observe a rapid water height increase during 2011–2014. After this period, water levels stabilize and slightly decrease but remain significantly higher than in 2009–2010. 2019 shows the highest water level for the Sobat river due to a renewed positive trend from 2017 onward. This suggests that the total SSWR IE was significantly higher

in 2018-2019 than the pre-2011 levels. In contrast, the IE data used in the process models, shown in Fig. 4a, have negative trends, which means that the process models do not account for the emission increase during 2010-2016 due to increasing IE, as suggested by Lunt et al. (2019).

IE estimates from the remote sensing-based SWAMPS also do not show the increase and underestimate annual mean IE.

Schroeder et al. (2015) have recommended not to use SWAMPS absolute IE as the microwave sensors used in SWAMPS have limited capability to detect water underneath the soil surface or beneath closed forest canopies. This effect can impact also the temporal IE changes, in addition to the absolute IE, as such changes beneath the forest canopies would also not be observed. It is unclear why TOPMODEL, which accounts for lateral water transport processes, does not capture the trend in river outflow. These are interesting topics for follow-on investigations.

**3.3.2 Seasonal cycle**

Next, we assess the seasonal cycle of the TROPOMI-derived emission estimates. Figure 5a shows the seasonal cycles in 2018 and 2019. The largest emissions are in DJF in 2018, while DJF, MAM and SON have large emissions of similar magnitude in 2019. In both 2018 and 2019, TROPOMI emissions are lowest in JJA; in contrast, the process models estimate the lowest emissions in DJF (Fig. 5c). We investigate this mismatch by looking at the seasonal cycle of IE in the models. The model

emissions have a strong correlation with their respective IE's (Wetcharts R = 0.91; LPJ-wsl R = 0.94, where R is correlation





coefficient), indicating that the seasonality of emissions is driven by IE. In fact, the differences in IE seasonality between LPJ-wsl and Wetcharts are consistent with the emissions differences; for example, both IE and emissions in LPJ-wsl are lower than in Wetcharts during MAM.

The seasonality of the altimetry-based river water height measurements, shown in Fig. 5d, is highest in SON and is very different from Wetcharts IE (highest in JJA). This can partially explain the difference in the seasonal cycles of Wetcharts and TROPOMI emissions. The seasonal cycle of Wetcharts Extended Ensemble IE shown here is strongly correlated with local precipitation (Fig. 5b), as the intra-annual IE variation is calculated using precipitation. However, this method would not accurately account for IE variation due to lateral water fluxes and evapotranspiration. Surface runoff is especially important
for river-fed wetlands like Sudd, whose IE is controlled by water inflow from the White Nile because regionally the evapotranspiration rate exceeds rainfall (Lunt et al., 2019; Sutcliffe and Brown, 2018). LPJ-wsl IE seasonality shows better agreement with the river height data as it is calculated using TOPMODEL, which accounts for the lateral fluxes and evapotranspiration. However, LPJ-wsl emissions still show large differences with the seasonal cycle of TROPOMI emissions. Previous remote sensing studies for the Sudd wetlands have found the largest IE during September-January in 2007-2008
(Robelo et al., 2012) and during December-January in 1991–1992 (Travaglia et al., 1995), in better agreement with river height measurements than the process models. Overall, IE seasonality of models appears to be significantly off, which can explain part of the mismatch between TROPOMI and model emissions.

    In both 2018 and 2019, TROPOMI emission estimates are the lowest during JJA, while river height measurements are the
lowest in MAM. A similar seasonal cycle mismatch in the GOSAT emission estimates and IE, derived using MODIS Land Surface Temperature (LST) as a proxy, is shown in Lunt et al (2019). They find the highest emissions trend during SON, which had the smallest trend in IE, but no trend in emissions during MAM, which has the highest IE trend (i.e. strongest negative LST trend).

An explanation for the difference in seasonal phasing can be a higher temperature dependence of emissions than suggested by the models as temperatures are lowest during JJA. We evaluate this hypothesis using Wetcharts Full Ensemble, which provides a total of 324 emission estimates for three temperature dependences q10 (=1, 2, 3; see Bloom et al., 2017). Figure 6 compares the average seasonal cycle of TROPOMI emissions with Wetcharts emissions using different q10's (see also Table 3). Wetcharts emissions with q10 = 1 have the poorest agreement with the seasonal cycle of TROPOMI (R = -0.71). Interestingly,
these emissions also have the lowest annual means (= 0.5 Tg yr$^{-1}$). Conversely, Wetcharts emissions with q10 = 3 have the best correlation with TROPOMI (R = -0.22) and have the largest annual mean (=1.0 Tg yr$^{-1}$). In fact, the member estimate— out of the 324-member Full Ensemble —with the largest annual emissions of 3.7 Tg yr$^{-1}$ also has the best correlation with TROPOMI (R = 0.13). As expected, this member uses q10 = 3. The agreement of TROPOMI with the larger q10 model estimates, in terms of both annual mean and seasonal cycle, suggests that wetland emissions from SSWR have a large





temperature dependence. In their study of wetlands in the Amazon Basin, Tunnicliffe et al. (2020) pointed to temperature as a more important control on methane emissions than IE. They find a simultaneous, spatially correlated emission and temperature increases in the west Brazilian Amazon during the El-Nino of 2015, with unchanged IE. Moreover, Wilson et al. (2016) found a negligible impact on wetlands emissions in the Amazon basin despite the large difference in precipitation between 2010 and 2011, which impacted IE significantly. Note that it is also possible that the higher q10's we find for SSWR emissions are

simply compensating for errors due to a remaining misrepresentation of IE, or other factors covarying with temperature.

Figure 7 shows the emission anomalies time series from TROPOMI along with temperature and IE, which we assume to be proportional to river height. A small lag between the river height and IE is expected, but we expect it to be negligible in comparison to a full season. We observe that the emissions show a strong correlation with temperature (R= 0.66), but a poor

correlation with IE (R= 0.07). The emissions peak a full season later than IE, and accounting for this seasonal lag improves the correlation significantly (R= 0.85). An explanation for this can be the higher temperature dependence of emissions discussed earlier. Another explanation could be the "activation" time of methanogenesis after flooding, as it takes time for anoxic conditions to develop and alternative electron acceptors to be depleted. Jerman et al. (2009) documented that methane emissions from water-saturated soil slurries remained very low for a long time: methane production started after a lag of 84

days at 15° C and a minimum of 7 days at 37° C, the optimum temperature for methanogenesis. They found that the lag was inversely related to iron reduction, which is expected as iron reduction outcompeted methanogenesis. Similarly, Itoh et al. (2011) investigated methane emissions from rice paddy fields and found a time lag of a few weeks between the onset of inundation and peak emissions.

Process models assume that wetland emissions are instantaneously regulated by IE, and they do not account for the time lag as information on the availability of alternate electron acceptors is generally not available. This results in an incorrect temporal allocation of the wetland emissions when the emissions are scaled with precipitation or even IE directly. For river floodplains like Sudd, scaling emissions directly with precipitation would give even worse estimates in models as the IE is mostly controlled by river inflow, and not the local precipitation, as the evapotranspiration rates exceed the rainfall in the region.


**5 Conclusions**

XCH$_4$ enhancements over South Sudan have been observed in remote sensing studies suggesting large emissions from the Sudd wetlands as the cause (Lunt et al., 2019, Hu et al., 2018, Frankenberg et al., 2011). We observe two large enhancements in the region in a 2-year average map of TROPOMI XCH$_4$—over Sudd, and Machar and Lotilla wetlands. Sudd Wetlands are

flooded by the White Nile river originating from Lake Victoria, while the wetlands in the east are around smaller rivers like the Sobat originating in the Ethiopian mountains. In this study, we examine these wetlands, and their river systems, together to understand the controls of the emissions causing the large XCH$_4$ enhancements.

We estimate methane emissions of $7.2 \pm 3.2$ Tg yr$^{-1}$ from wetlands in South Sudan during 2018–2019 using a mass balance
approach applied to TROPOMI data. We find large differences between the emission estimates from TROPOMI and wetland
process models LPJ-wsl and Wetcharts. The annual mean estimates from TROPOMI are an order of magnitude larger than
from the models, which may be explained by the up to 4 times underestimated IE in the models. We find differences in
interannual variability and average seasonal cycles of TROPOMI and models, which can be again, partially explained by the
strong dependence of model emissions on poor IE estimates. We find that the Wetcharts emission estimates that use high-
temperature dependence (q10 = 3) show a better agreement with TROPOMI concerning both seasonality and annual emissions.
This indicates that the models may also underestimate the temperature sensitivity of the methane emissions. The causes of this
need to be investigated further.

The IE of SSWR is analyzed using satellite altimetry-based river height measurements of White Nile and Sobat rivers at
locations within the Sudd and Macher wetlands. The IE estimates used in models are based on the local precipitation, whereas,
the actual IE of SSWR is driven by water inflow from the rivers as evapotranspiration exceeds the local precipitation. As a
result, both the seasonal cycle and trend of model IE disagree with river height data. The seasonal cycle of IE from river height
data shows better agreement with the TROPOMI emissions when a full season-long lag between the two is assumed. This time
lag can be explained by the time needed for methanogenesis to develop in the seasonally flooded areas of the wetlands. A more
precise estimate of the lag is not possible due to the coarse temporal resolution of our TROPOMI emissions estimates.

The lack of information on the correct relationship of wetland emissions with IE and temperature results in large model
uncertainties. Such large gaps in our understanding of the process driving wetland emissions call for further investigation. As
shown here for the wetlands of South Sudan, TROPOMI provides valuable observations over remote and inaccessible wetland
regions of the world, which future wetland studies can take advantage of.

**APPENDIX**

**Section A1. Systematic Measurement Uncertainties**

Surface albedo and aerosols can alter the optical light path, introducing biases in XCH$_4$ (Butz et al., 2011). Therefore, the
XCH$_4$ enhancement over South Sudan can be affected by the differences between the source and background region values of
these parameters. The average retrieved aerosol optical thickness (AOT) and surface albedo in the SWIR band of TROPOMI
are shown in Fig. A1. For SSWR and its background, the AOT and albedo differences in two-year average data are 0.001 and
-0.10, respectively. The average differences for seasonal average maps are $-0.01 \pm 0.01$, $-0.15 \pm 0.02$ and $16.3 \pm 8.4$ ppb for
AOT, albedo and XCH$_4$ respectively. The negative albedo difference for SSWR occurs due to the high albedo the Sahara in

the background. This small albedo difference is unlikely to influence the SSWR XCH4 enhancement significantly, especially, as an albedo-based bias correction is applied to the XCH4 in operational TROPOMI product (Hasekamp et al., 2019). We also examined the possibility that the XCH4 enhancement over South Sudan is an artefact of sun glint geometry of TROPOMI observations due to refection on standing water of the Lakes and inundated areas in the region. This can happen when the observation geometry over a water body surface is at the specular reflection angle, the viewing zenith angle matches the solar

zenith angle, causing a spike in the level 1 radiance measurements. However, this was found not to occur over the wetlands of South Sudan.

*Data Availability*. TROPOMI data are available at the Copernicus Open Access Hub (https://scihub.copernicus.eu/)

Satellite altimetry river height dataset is available at Hydroweb website (http://hydroweb.theia-land.fr/). Wetcharts data can be downloaded from https://daac.ornl.gov/CMS/guides/CMS_Global_Monthly_Wetland_CH4.html. SWAMPS IE data is available at http://wetlands.jpl.nasa.gov. LPJ-wsl data is available from Zhen Zhang (yuisheng@gmail.com) upon request.

*Author contributions*. The study was designed by SP, IA & SH. AL and TB provided the TROPOMI XCH4 data. AAB provided

Wetcharts data. BP and ZZ provided LPJ-wsl data. SP and MT performed the analysis. SP, IA & SH wrote the manuscript using contributions from all the co-authors.

*Acknowledgements*. We thank the team that has realized the TROPOMI instrument, consisting of the partnership between Airbus Defence and Space Netherlands, Royal Netherlands Meteorological Institute (KNMI), SRON Netherlands Institute for

Space Research (TNO), and Netherlands Organisation for Applied Scientific Research (TNO), Netherlands Space Office (NSO), and European Space Agency. This research is supported through the GALES (Gas Leaks from Space) project (Grant 15597) by the Dutch Technology Foundation, which is part of the Netherlands Organisation for Scientific Research (NWO), and is partly funded by Ministry of Economic Affairs, The Netherlands. Part of this research was conducted at the Jet Propulsion Laboratory, California Institute of Technology, Anthony Bloom acknowledges support from NASA Earth Science

grant (#NNH14ZDA001N-CMS). Benjamin Poulter and Zhen Zhang acknowledge support from the Gordon and Betty Moore Foundation through Grant GBMF5439 'Advancing Understanding of the Global Methane Cycle' supporting the Methane Budget activity for the Global Carbon Project (globalcarbonproject.org)

*Competing interests*. The authors declare that they have no competing interests.






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





**Table 1.** SSWR XCH₄ enhancement and emission estimates. The enhancements ($\Delta XCH_4$) is the XCH₄ difference between
SSWR and the background. Data coverage is the fraction of the number of $0.1° \times 0.1°$ grid cells in SSWR with TROPOMI
data. Wind speed is the average boundary layer winds from ERA5. Emission estimates are calculated using Eq. (1). $\pm$
represents 1 $\sigma$ uncertainty.

| Season | Data coverage (%) | Wind speed (m s⁻¹) | XCH₄ enhancement (ppb) | Emissions (Tg yr⁻¹) |
|--------|-------------------|--------------------|------------------------|---------------------|
| DJF-2018 | 91 | 3.5 ± 0.9 | 21.7 + 1.5 | 15.1 ± 5.2 |
| MAM-2018 | 74 | 2.1 ± 1.2 | 21.6 ± 1.4 | 9.0 ± 5.3 |
| JJA-2018 | 44 | 2.7 ± 0.3 | 7.3 ± 2.4 | 4.0 ± 2.0 |
| SON-2018 | 83 | 2.7 ± 0.4 | 16.5 ± 2.0 | 8.9 ± 2.0 |
| DJF-2019 | 98 | 3.2 ± 0.9 | 14.4 ± 1.1 | 9.2 ±2.7 |
| MAM-2019 | 92 | 3.1 ± 0.8 | 14.8 ± 1.2 | 9.1 ±3.0 |
| JJA-2019 | 43 | 2.8 ± 0.3 | -1.5 ± 2.4 | -0.9 ±1.4 |
| SON-2019 | 83 | 1.9 ± 0.2 | 26.3 ± 2.2 | 9.8 ±1.4 |






**Table 2.** Annual emission and IE estimates for SSWR. The values in parentheses show 1-standard deviation spread over the given periods. The dashed values give the range of Wetcharts ensemble estimates.


| | Period | Maximum IE ($10^3$ km$^2$) | Emissions (Tg yr$^{-1}$) |
|---|---|---|---|
| Wetcharts Extended Ensemble/GLWD | 2001-2015 | 32 (7) | 0.4 (0.1) – 1.0 (0.2) |
| Wetcharts Extended Ensemble/GLOBCOVER | 2001-2015 | 69 (10) | 0.70 (0.1) – 1.8 (0.2) |
| Wetcharts Full Ensemble | 2010 | 30–66 | 0.07 – 3.7 |
| LPJ-wsl/ TOPMODEL | 1980-2016 | 57 (9) | 1.1 (0.25) |
| SWAMPS | 2001-2019 | 25 (5) | – |
| Gumbricht et al. (2017) | 2011 | 99 | – |
| TROPOMI | 2018-2019 | – | 7.2 ± 3.2 |





**Table 3.** Annual emission estimates from Wetcharts Full ensemble (2010) for different temperature dependencies, and
correlation coefficient (R) of their respective average seasonal cycle with TROPOMI emissions.

| Temperature dependence (q10) | Emissions (Tg yr$^{-1}$) | R (with TROPOMI) |
|---|---|---|
| 1 | 0.5 | -0.71 |
| 2 | 0.8 | -0.40 |
| 3 | 1.0 | -0.22 |
| Maximum* | 3.7 | 0.13 |

*The maximum annual emission estimate in the 324-member ensemble of Wetcharts Full Ensemble. Its q10 is 3.





**Table A1.** The total permanent SSWR IE from Huges & Huges (1992).


| Wetlands in SSWR | Wetland Area (km$^2$) |
|---|---|
| Sudd | 16,500 |
| Machar marshes | 9,000 |
| Lottila Swamps | 2,000 |
| Veveno, Adiet and LiLebook | 6,500 |
| Kenamuke and Kobowen swamps | 1,700 |
| Bahr el Ghazal floodplains | 900 |
| Total | 36,000 |


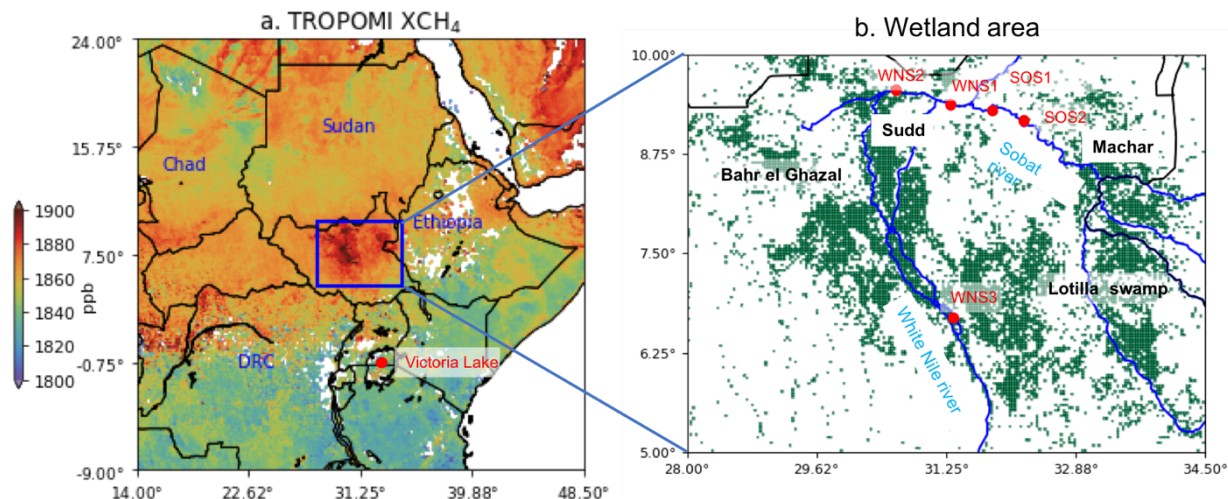


**Figure 1.** The TROPOMI XCH$_4$ enhancement over the South Sudan wetlands. (a) Average of two years (December 2017 to November 2019) of TROPOMI XCH$_4$ at 0.1° × 0.1° resolution (b) Wetlands in South Sudan from Gumbricht et al. (2017) are shown in green, and the rivers in the region are shown in blue. The area within the blue rectangle (5°–10° N and 28°–34.5° E) is referred to as South Sudan wetlands region (SSWR). The red dots show the locations of satellite altimetry-based river water

height measurement sites.

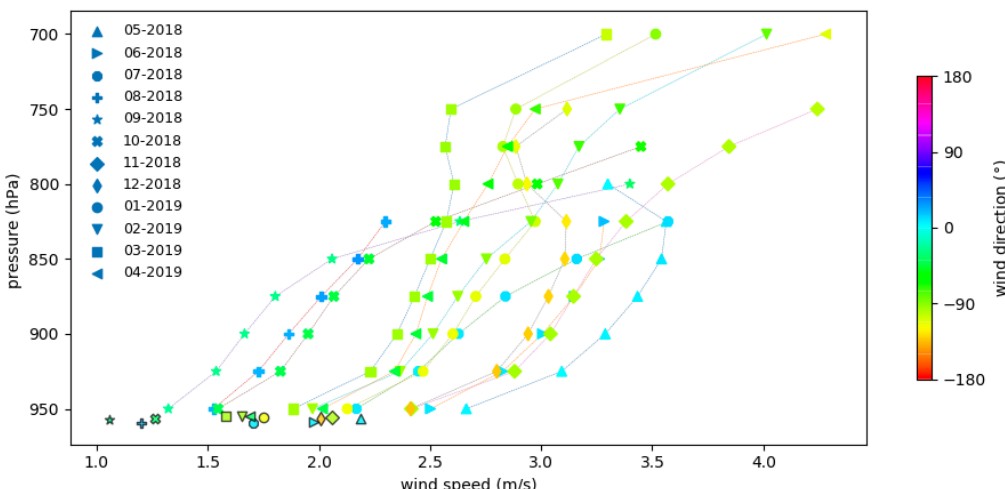


**Figure 2.** Monthly average boundary layer ERA5 winds in SSWR. Wind speeds (X-axis) and directions (colour of the markers) at 10:00 UTC, which is the closest hour to the local TROPOMI overpass time, are shown at different pressure levels of the model. The markers with dark edges represent 10-meter height winds.



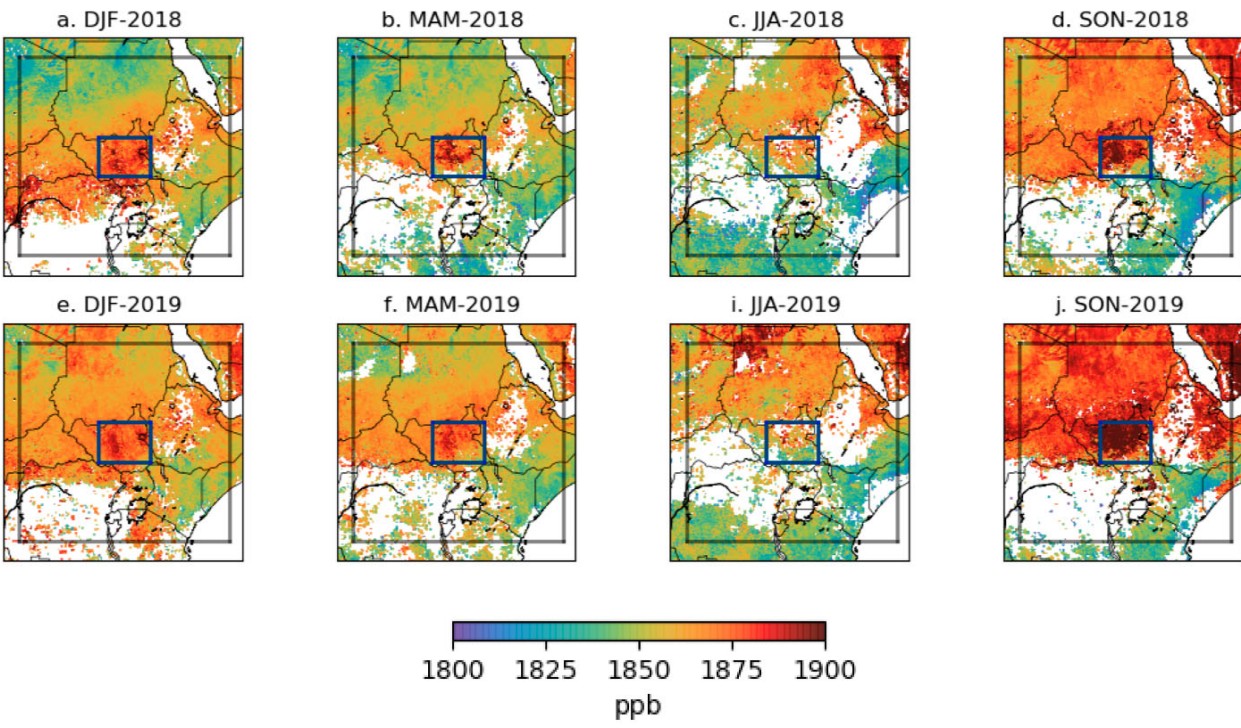

**Figure 3.** Seasonally average TROPOMI XCH$_4$ at 0.1° × 0.1° resolution. The blue rectangles show SSWR and the larger black rectangles show the background areas used for calculating XCH$_4$ enhancements. Note that DJF of a year includes December of the previous year.

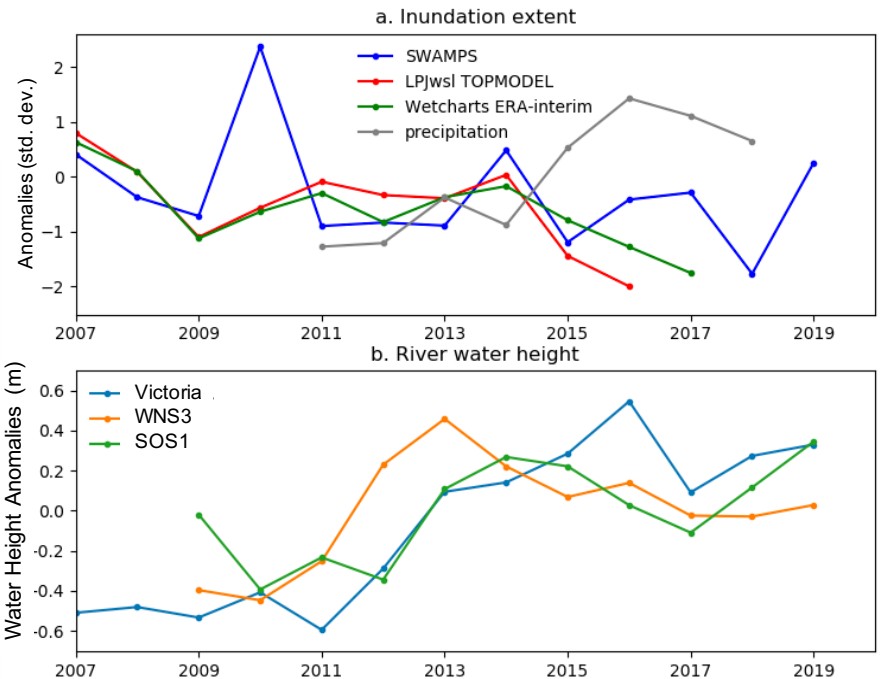

**Figure 4.** SSWR IE estimates. (a) Annual anomalies of IE estimates of SWAMPS, TOPMODEL and Wetcharts (ERA-interim), used in process models, and ERA5 precipitation, expressed in the unit of the standard deviation of the respective annualized time series. (b) Water height anomalies for the altimetry sites in Lake Victoria, and White Nile (WNS3) and Sobat (SOS1) rivers from the Hydroweb database. Locations of these altimetry sites are given in Figure 1. Here we only use the altimetry sites which have a sufficiently long temporal coverage that includes 2018-2019.




**Figure 5.** Mean seasonal cycles expressed in the unit of the standard deviation of respective time series. (a) Emission estimates from TROPOMI; (b) precipitation and temperature from ERA5 (2010-2019); (c) emissions and IE from the process models LPJ-wsl and Wetcharts Extended Ensemble; (d) river water height measurements at the altimetry sites given in Figure 1b. The
vertical bars represent 1-standard deviation spread over different years.





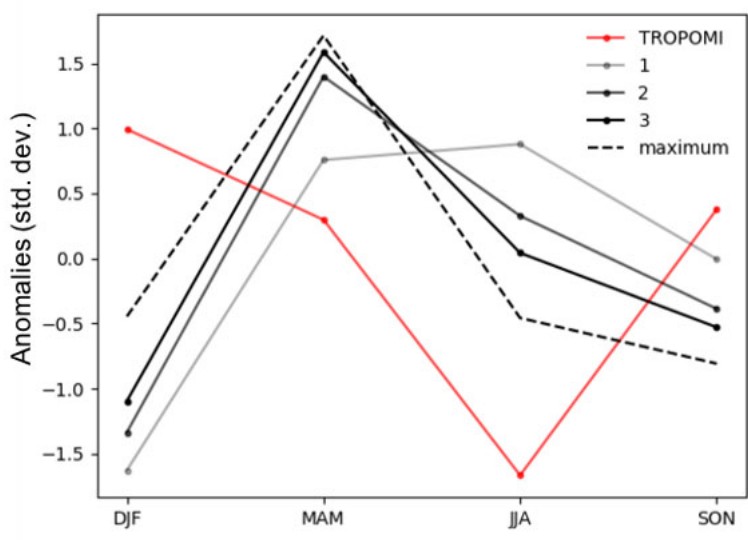

**Figure 6.** Seasonal cycle of SSWR emissions. Methane emissions from Wetcharts Full Ensemble for 2010 (December 2009 – November 2010) and TROPOMI are shown. The solid lines show the average of an emission estimate ensemble for a temperature dependency (q10). The dashed line shows the seasonal cycle of the Wetcharts estimate that has the largest annual emissions. All values are shown in the unit of standard deviation.



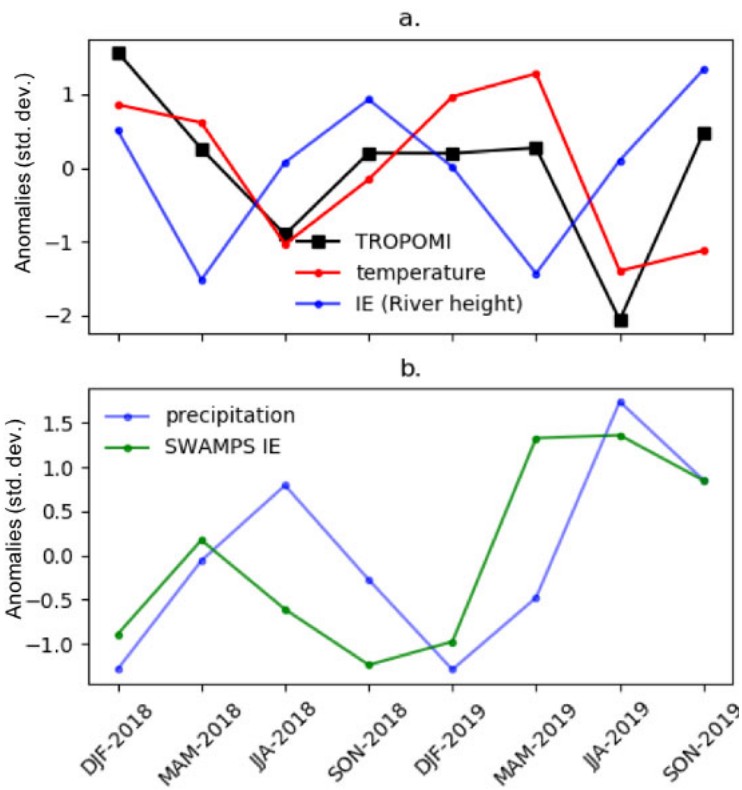

**Figure 7.** Seasonal anomalies in SSWR. (a) emissions estimates of TROPOMI, ERA5 temperature and IE using river height
measurements are shown. (b) local precipitation and SWAMPS IE data are shown. All values are expressed in the unit of
standard deviation. Correlation coefficients (R) of TROPOMI emissions with temperature, river height, SWAMPS and
precipitation are 0.66 and 0.07, –0.41, –0.80, respectively.





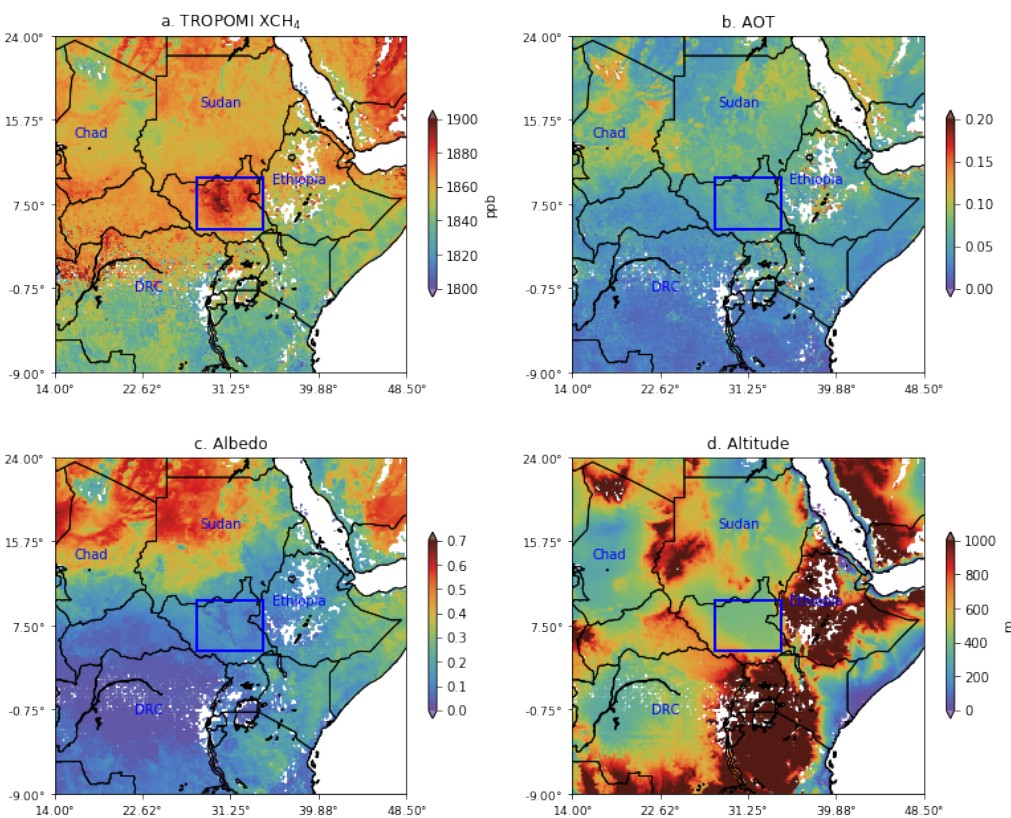

**Figure A1.** Average TROPOMI data (2018-2019) at 0.1° × 0.1° resolution. The albedo and AOT are retrieved for the SWIR
band at 2.3 $\mu$m.