# Peer review of "Using satellite data to identify the methane emission controls of South Sudan's wetlands"

_Biogeosciences, 2020_

## Referee Comment (RC1) · Anonymous Referee #1 · 26 Aug 2020

General

This is a thoughtful and well-argued paper on an exceptionally important problem, that should certainly be published. It has much fresh insight, and makes innovative use of the powerful new data from the TROPOMI instrument. However it has some significant shortcomings that need to be considered before the study is accepted.

The atmospheric methane burden is rising, and a significant part of that growth appears to come from the northern tropics. In this context, South Sudan has some of the world's most important wetlands, primarily in the Sudd swamps of the White Nile river, and also in other wetlands along its tributary Sobat river, from Ethiopia. Secondly, it is not clear if the current rise in methane arises directly from human actions that increase emissions, or as a warming-feeds-warming feedback from global warming.

[Figure]

Thus accurate quantification of South Sudan's emissions is globally important, and it is also a major question how these emissions will respond to changes in both temperature and inundation. But South Sudan is relatively inaccessible to sustained scientific field work as there are violent conflicts in the region, in the South Sudanese Civil War (2013-2020).

Pandey et al. address the task of quantifying South Sudan's methane emission by using TROPOMI observations from the S-5P satellite. This is immensely valuable in global terms – it is a key area – and the paper makes a major and vital contribution.

I do however have a number of specific concerns.

1. Cattle. The general concern is that cattle and other ruminants are not mentioned at all. South Sudan has one of the world's highest 'per-human capita' ruminant populations. It is hard to get good numbers, but old FAO data suggest 12 million cows, 14 million goats and 13 million sheep. I have not myself been to South Sudan, though I have visited both north Sudan and Uganda: cattle are everywhere. So are goats. Moreover, cattle and swamps go together – some African swamps have cows on every bit of available footing. This is ruminant heaven. This is not a minor matter – there is much debate whether tropical methane emissions are rising because swamp methanogenesis is increasing the direct emission of methane to the air (e.g. via plant stems, or ebullition), or whether the methane emissions are rising because of increasing cattle populations (e.g. see Schaefer et al. Science 2016 and Nisbet et al. G.B.C. 2016,2019). If the main emitter is the swamp directly, then flux will depend on temperature and inundation. Global warming will feed back into more emission. It will be very hard to mitigate this locally, except by cutting warming globally. But if the main methane emitters are the cattle that eat the swamp vegetation, then emissions may also be partly dependent on cattle numbers – the growth in emissions will in part come from human actions in increasing cattle populations. In principle, then, emissions can be mitigated by reducing cattle populations. In South Sudan this is particularly relevant as the cattle populations are not simply food producers but also held as 'currency'. There are far

more cattle than needed for food. Switching to easier-to-maintain non-cattle 'money' (e.g. cellphone accounts, as in neighbouring Uganda) can lead to reduction in cattle numbers and hence cut methane emissions. It would be a large cultural shift, but feasible and would have significant impact in mitigating emissions – if the emissions come from the cows, and are thus subject to human intervention. To sum up – surely cows deserve a mention!! – after all, there is discussion of waste water management! Also manure management is an odd factor here – where is that done in South Sudan? This is a very different place from the US with its manure lagoons and industrial cattle. My experience of the region is that plops land where plops land. They may be picked up to burn as biofuel in winter, but there is little "management".

2. Mass balance approach. To assess fluxes, the paper depends on the work of Buchwitz et al. 2017. This is interesting and innovative, but the Buchwitz et al. approach focussed on hotspots like the Four Corners region. Although that paper did go to the more regional scale of Turkmenistan, the scale on which the method is being applied in this paper is significantly more than the main Buchwitz et al applications. Also, the region is perhaps rather inhomogeneous with regard to distribution of methane sources.

3. I note that in some quarterly periods, the 'background' region (fig 3) crosses the 'methane waterfall' at the Inter-Tropical Convergence Zone, with a major difference of maybe circa 30ppb between values 100km north and 100km south of the ITCZ. Is this valid for background? - Methane in the Southern Hemisphere is MUCH lower. The situation is complex, because the ITCZ does reach South Sudan in the northern summer, but in the equinoctial months it is close to Uganda. Thus, given the proximity of the ITCZ, and the inhomogeneity of the region, and I'd imagine defining a background could be very difficult, yet crucial to the modelling. Note that errors in JJA (wet season) are large compared to fluxes and data coverage low. The result of -1.5±2.4 ppb enhancement looks very odd at the peak of the rains. Could this be a background effect with the much-lower CH4 Southern Hemisphere air so close by, south of the ITCZ? Are the conclusions justified given this?

4. The study only uses observations under cloud-free conditions. That's obvious – or is it? In the wet season under the Inter-Tropical Convergence Zone, cloud-free means that a major change has occurred – the ITCZ has moved, and winds and advection are different. It's an almost insoluble problem for satellite observation: the action is under the clouds. However, there are period – for example mornings, or occasional days when thunderstorm cells are present but not in the immediate area – when the ITCZ is present but clouds absent. But these low advection periods might be cooler or otherwise different than average cloudy days. Thus it would help to discuss this problem – are cloud-free conditions valid conditions to determine 'typical' fluxes?

5. Line 316. JJA and Temperature dependence. This is a very interesting finding and probably should be emphasised more in the abstract and conclusion, as the implications for emission modelling are important. As the authors correctly point out, in this region (e.g. Malakal), the summer months July and August are the coolest months of the year – because of the cloud cover. Pre-rains March and August are hottest. Incidentally, are temperatures from ECMWF (line 135)? Or are they checked against local values, for example from Malakal? Sometimes ECMWF can be not good in recognising local factors.

There is a typo in Table 1 top line – XCH4 is plus/minus, not simply plus as written. A minor point – 'IE' for inundation extent, is an acronym too far. Why not just say 'inundation extent' each time and save us the reference to Ireland every time.

The manuscript is otherwise well written and well presented.

To conclude, this is an important paper that should be accepted after a re-think and minor revision. I would suggest amendments, particularly to discuss cattle, in this extremely cow-focussed region, and also to consider issues around defining the background.

---

## Referee Comment (RC2) · Anonymous Referee #2 · 14 Sep 2020

Summary and general comments:

Pandey et al. analyse recent observations of XCH4 over South Sudan to infer wetland emissions resolved at the seasonal time scale. They compare their findings with different process-based wetland models and report significantly higher emission estimates using a mass-balance approach. With additional data and assumptions they infer that inundation extend and or temperature sensitivity are a likely cause of this top-down versus bottom-up discrepancy and suggest that satellite studies, such as this, are useful tools for studying remote or hard to access areas. The paper is very well written, clearly structured and easy to read. The topic is of significant interest to the wider community and is a nice case study of potential ways to exploit TROPOMI data. However, there are two key issues (below) the authors need to address before this study should

[Figure]

be considered for publication in Biogeosciences.

1.) Methane emissions from ruminants in South Sudan

The authors use an emission estimate from EDGAR V4.3.2 of 0.36 Tg yr-1. Firstly, there is no real discussion of uncertainties of this number, although studies in other countries have shown that significant differences between top-down and bottom-up based estimates exist, even in countries with long-standing dedicated infrastructure for tracking livestock head counts, which might not be the case for a more recently formed country, such as South Sudan. For example, Miller et al. 2013 (https://www.pnas.org/content/110/50/20018) have claimed that agricultural emissions are severely underestimated in the US. As the proposed domain nearly covers all of South Sudan we can try a back of the envelope calculation on what national ruminant emissions could be. According to an FAO webpage (not the ideal source, http://www.fao.org/emergencies/fao-in-action/stories/stories-detail/en/c/326186/) there are 12M cattle, 20M sheep and 25M goats in South Sudan Assuming: 5 kgCH4 yr-1 for goats&sheep and 35kCH4 yr-1 for cattle in developing countries this yields ca. 0.65Tg CH4, so nearly twice the EDGAR V4.3.2 estimate. Emission factors from https://www.tandfonline.com/doi/pdf/10.3402/tellusb.v38i3-4.15135 It seems logical to address this issue in more detail in section 3.2.

2.) Discussion of impacts of seasonally changing spatial coverage and/or clear sky bias of observations.

The authors should further expand on the issue of data coverage and (potential?) clear-sky bias in the manuscript. Is this a significant source of uncertainty and how was this accounted for. The seasonal decrease in emissions in JJA coincides with a significant drop in data coverage. Emissions in JJA-2019 are reported to be miniscule (statistically close to 0), but this is not appropriately addressed in the manuscript. What is happening here? Assuming that agricultural and other methane sources are still active in JJA-2019 this result is even more extraordinary.

Specific comments:

Line 54: Please clarify: 38-56% higher than which RCP? They differ quite significantly in their anthropogenic emissions.

Line 105: Why do you choose to use Wetcharts data for 2009-2010 as basis for comparison here, although previous work you referenced (line 70) has already shown a strong trend in CH4 emissions in the Sudd wetlands after 2010, due to IE expansion?

L109: The wetland extent datasets (Lehner and Doell 2004, and Bontemps et al. 2011) seem not to be very up to date and able to include any trends happening after 2010.

L170: What is the temporal resolution of the meteo data set used here? Are the wind speeds from these 4 consecutive hours really independent or is the model only constrained at lower frequency intervals? This could lead to an artificially low variability. Are there any surface observations in the (wider) region to compare with the model?

Line 233: Why are emissions now 7+/-3.2 Tg yr-1 and not 7.2+/-3.2 Tg yr-1 ?

Line 239: Why is the TROPOMI-based wetland CH4 emission estimate compared to a minor source such as oil and gas emissions (0.05 Tg yr-1) here? The biggest anthropogenic source in the region, even according to EDGAR, is agriculture at 0.36 Tg yr-1, which is an order of magnitude bigger than O&G (possibly bigger, see general comments).

Line 246: Please correct to 'nearly an order of magnitude lower'.

Line 252: It seems the upper end of the WetCharts ensemble is not that far off the lower end of your estimate here, so claiming they differ by an order of magnitude seems unhelpful.

Line 287: JJA-2019 seems extremely low, this needs to be discussed further (see general comments).

Line 297: It is unclear if TROPOMI-derived CH4 emission estimates really track river

water heights all that well or if the effect is mostly Q10 related. Maybe those two components should not be singled out or it should be made clear early in the manuscript that both components contribute.

Line 316: Temperatures are surely lower in JJA, but not low enough to explain the emissions reported for JJA-2019.

Line 362: Some models report emissions that are definitely NOT 'an order of magnitude' smaller.

Line 365: One model ensemble member with Q10 = 3 estimates 3.7 Tg yr-1 despite the 'poor IE estimates'. So it seems this study cannot disentangle the two issues, which should be reflected in the conclusion here. Furthermore, after reading this manuscript it seems more convincing that Q10 is the key issue rather than IE.

Table 1: Please elaborate what the variable 'data coverage' signifies here. For example, for DJF-2018: is it 91% of all cells were covered at least once in this 3 month period or were 91% of all possible data collected for an average cell. For example, if you measure every cell at least once in DJF-2018 would you label this as 100% coverage (which it is not) or do you need to measure all cells, all of the time (once per day) to reach 100% coverage?

---

## Author Comment (AC1) · 7 Oct 2020

We thank the anonymous referee for his time and useful feedback, which helped further improve the paper. Our point-wise responses to the referee's comments (in *italics*) are as follows:

**Referee**: *1. Cattle. The general concern is that cattle and other ruminants are not mentioned at all. South Sudan has one of the world's highest 'per-human capita' ruminant populations. It is hard to get good numbers, but old FAO data suggest 12 million cows, 14 million goats and 13 million sheep. I have not myself been to South Sudan, though I have visited both north Sudan and Uganda: cattle are everywhere. So are*

*goats. Moreover, cattle and swamps go together – some African swamps have cows on every bit of available footing. This is ruminant heaven. This is not a minor matter – there is much debate whether tropical methane emissions are rising because swamp methanogenesis is increasing the direct emission of methane to the air (e.g. via plant stems, or ebullition), or whether the methane emissions are rising because of increasing cattle populations (e.g. see Schaefer et al. Science 2016 and Nisbet et al. G.B.C. 2016,2019). If the main emitter is the swamp directly, then flux will depend on temperature and inundation. Global warming will feed back into more emission. It will be very hard to mitigate this locally, except by cutting warming globally. But if the main methane emitters are the cattle that eat the swamp vegetation, then emissions may also be partly dependent on cattle numbers – the growth in emissions will in part come from human actions in increasing cattle populations. In principle, then, emissions can be mitigated by reducing cattle populations. In South Sudan this is particularly relevant as the cattle populations are not simply food producers but also held as 'currency'. There are far more cattle than needed for food. Switching to easier-to-maintain non-cattle 'money' (e.g. cellphone accounts, as in neighbouring Uganda) can lead to reduction in cattle numbers and hence cut methane emissions. It would be a large cultural shift, but feasible and would have significant impact in mitigating emissions – if the emissions come from the cows, and are thus subject to human intervention. To sum up – surely cows deserve a mention!! – after all, there is discussion of waste water management! Also manure management is an odd factor here – where is that done in South Sudan? This is a very different place from the US with its manure lagoons and industrial cattle. My experience of the region is that plops land where plops land. They may be picked up to burn as biofuel in winter, but there is little "management".*

**Authors**: We agree with the referee that livestock is an important contributor to the methane emissions from South Sudan, and it needs to be addressed in our paper. We will add the following text to the revised manuscript:

"Livestock is the largest anthropogenic methane source in SSWR region: 0.36 Tg yr$^{-1}$ in 2012 as per EDGAR v4.3.2. In 2015, it had increased to 0.37 Tg yr$^{-1}$ as per EDGAR

v5 (Crippa et al., 2020). South Sudan has a large population of livestock: 7.5 million dairy cattle, 4.6 million non-dairy cattle, 13.5 million goats and 16.3 million sheep in 2018, which causes 0.63 Tg yr$^{-1}$ of methane emissions (FAOSTATS., 2020). This amount is twice of what we use to calculate the wetlands emissions for SSWR. In the extreme case that all these additional emissions are located in SSWR, it would slightly reduce our wetland emission estimate, however, well within its uncertainty margin."

**Referee**: *2. Mass balance approach. To assess fluxes, the paper depends on the work of Buchwitz et al. 2017. This is interesting and innovative, but the Buchwitz et al. approach focussed on hotspots like the Four Corners region. Although that paper did go to the more regional scale of Turkmenistan, the scale on which the method is being applied in this paper is significantly more than the main Buchwitz et al applications. Also, the region is perhaps rather inhomogeneous with regard to distribution of methane sources.*

**Authors**: Mass balance approaches have been used for emission quantification using TROPOMI XCH4 data in multiple recent studies (Pandey et al., 2019; Schneising et al., 2020; Varon et al., 2019; Zhang., et al 2020). Pandey et al. (2019) and Varon at al. (2019) apply the CSF (Cross-Sectional Flux) mass balance approach to methane plumes visible in individual overpasses of TROPOMI. Schneising et al. (2020) apply a Gaussian integral mass balance approach to individual overpasses of TROPOMI over the large Permian oil and gas basin of USA and find emission estimates consistent with GEOS-chem inversion results of Zhang et al (2020). The mass balance approach of Buchwitz et al. (2017) is also used in Zhang et al. (2020) to quantify emissions from the Permian basin region with effective length ($L = area^{0.5}$) of 375 km, which yields emission estimates consistent with the GEOS-Chem inversion of the study. The $L$ of our South Sudan wetland region (SSWR) is 632 km. The size of the background for the Permian study was $24 \times 24$ degrees latitude-longitude, comparable to our background of 25 $\times$ 26.5 degrees latitude-longitude. Further, Buchwitz et al. (2017) successfully test their approach on some large regions: Turkmenistan ($L = 700$km) and Azerbaijan

($L = 300$km). In addition, our emission estimates are agreement with that of GOSAT inversion emissions of Lunt et al. (2019) for the latest year in their inversion (5.2–6.9 Tg yr$^{-1}$ for 2016), especially when the source region differences between the two studies are accounted for. We expect the impact of emissions inhomogeneity to be small and within the uncertainty margin of the quarterly emission estimates. Overall, the performance of the method has proven to be sufficiently accurate for the purpose of this study.

**Referee**: *3. I note that in some quarterly periods, the 'background' region (fig 3) crosses the 'methane waterfall' at the Inter-Tropical Convergence Zone, with a major difference of maybe circa 30 ppb between values 100km north and 100km south of the ITCZ. Is this valid for background? - Methane in the Southern Hemisphere is MUCH lower. The situation is complex, because the ITCZ does reach South Sudan in the northern summer, but in the equinoctial months it is close to Uganda. Thus, given the proximity of the ITCZ, and the inhomogeneity of the region, and I'd imagine defining a background could be very difficult, yet crucial to the modelling. Note that errors in JJA (wet season) are large compared to fluxes and data coverage low. The result of -1.5 ± 2.4 ppb enhancement looks very odd at the peak of the rains. Could this be a background effect with the much-lower CH4 Southern Hemisphere air so close by, south of the ITCZ? Are the conclusions justified given this?*

**Authors**: We thanks the reviewer for pointing this out, and we agree that the position of the ITCZ in different seasons can give rise to biases in our estimates. To address this, we have updated emission estimation method. Now, we first subtract a 3rd order polynomial fit of XCH4 as a function of latitude from the XCH4 maps used for emission quantification. This accounts for the influence of the methane "water fall" along the ITCZ on the background. To derive the polynomial fit, we exclude the longitudes crossing the SSWR region to ensure the fit is not affected by the source region emissions. The attached figure shows averages quarterly XCH4 plots after removing the latitudinal fits (it will replace Figure 3 of the manuscript). We find a new total emission estimate

of $8.2 \pm 3.2$ Tg yr$^{-1}$ (old estimate: $8.0 \pm 3.2$ Tg yr -1 ) and wetland emission estimate from SSWR to be $7.4 \pm 3.2$ Tg yr$^{-1}$, in statistical agreement with our previous estimate of $7.2 \pm 3.2$ Tg yr$^{-1}$. The following text will be added to the revised manuscript:

"We remove the large scale latitudinal XCH4 gradient from the seasonal average TROPOMI XCH4 maps by subtracting a 3rd order polynomial fit from the background region, excluding the source region (see Figure 2)."

"Table 1. SSWR XCH4 enhancement and emission estimates. The enhancements ($\Delta$ XCH4 ) is the XCH4 difference between SSWR and the background after correcting XCH4 for latitudinal variation. Data coverage is defined as the fraction of $0.1 \times 0.1$ degrees grid cells in SSWR with at least five days of high-quality TROPOMI measurements in a quarter. Wind speed is the average boundary layer wind from ERA5. Emission estimates are calculated using Eq.(1). $\pm$ represents the $1\sigma$ uncertainty." (see supplement )

"Figure 2. Seasonal average TROPOMI XCH4 (ppb) at $0.1 \times 0.1$ degree resolution. The black rectangles at the centre of each panel show the SSWR source region. The area outside of it is used as a background. XCH4 is corrected for large-scale latitudinal variation by subtracting a 3rd order polynomial fit using the region shown by blue rectangles, which excludes the longitudes of the source region. Note that DJF includes December of the previous year." (see supplement)

**Referee**: *4. The study only uses observations under cloud-free conditions. That's obvious – or is it? In the wet season under the Inter-Tropical Convergence Zone, cloud-free means that a major change has occurred – the ITCZ has moved, and winds and advection are different. It's an almost insoluble problem for satellite observation: the action is under the clouds. However, there are period – for example mornings, or occasional days when thunderstorm cells are present but not in the immediate area – when the ITCZ is present but clouds absent. But these low advection periods might be cooler or otherwise different than average cloudy days. Thus it would help to discuss*

*this problem – are cloud-free conditions valid conditions to determine 'typical' fluxes?*

**Authors**: Yes, we only use cloud-free observations, selected using the highest quality filter (*qa* = 1 ) which masks the presence of clouds and cirrus. This causes a sampling bias, but it will only affect our estimates of the emissions influenced by cloud cover. To our knowledge, there is no direct evidence that such a relationship exists, although we cannot exclude the possibility that it has some influence. TROPOMI provides enough measurements to sample the region throughout the year, which we consider the most important.

**Referee**: *5. Line 316. JJA and Temperature dependence. This is a very interesting finding and probably should be emphasised more in the abstract and conclusion, as the implications for emission modelling are important. As the authors correctly point out, in this region (e.g. Malakal), the summer months July and August are the coolest months of the year – because of the cloud cover. Pre-rains March and August are hottest. Incidentally, are temperatures from ECMWF (line 135)? Or are they checked against local values, for example from Malakal? Sometimes ECMWF can be not good in recognising local factors.*

**Authors**: Temperatures are taken from ECMWF ERA5. The seasonality of this data is consistent with average weather in Malakal (https://weatherspark.com/y/96893/Average-Weather-in-Malakal-South-Sudan-Year-Round,accessed1-10-2020). We will add the following text to the revised manuscript to highlight the JJA and temperature dependence finding.

To abstract:

"We find the lowest emission in the highest precipitation and lowest temperature season JJA, when models estimate large emissions. In general, our emission estimates show better agreement, in terms of both seasonal cycle and annual mean, with model estimates that use a stronger temperature dependence. This suggests that temperature might be a stronger control for the South Sudan wetlands emissions than currently assumed by models."

To conclusions:

"We find the lowest emissions in the highest precipitation and lowest temperature season of JJA, when models estimate large emissions as they assume a strong influence of the precipitation-derived inundation extent. We find that the Wetcharts emission estimates that use high-temperature dependence ($q10 = 3$) show a better agreement with TROPOMI concerning both seasonality and annual emissions. This indicates that the models may also underestimate the temperature sensitivity of the methane emissions. The causes of this need to be investigated further."

**Referee**: *There is a typo in Table 1 top line – XCH4 is plus/minus, not simply plus as written. A minor point – 'IE' for inundation extent, is an acronym too far. Why not just say 'inundation extent' each time and save us the reference to Ireland every time.*

**Authors**: Done

**Referee**: *The manuscript is otherwise well written and well presented. To conclude, this is an important paper that should be accepted after a re-think and minor revision. I would suggest amendments, particularly to discuss cattle, in this extremely cow-focussed region, and also to consider issues around defining the background.*

**References**: (In addition to the manuscript references)

- Crippa, M., Solazzo, E., Huang, G. et al. High resolution temporal profiles in the Emissions Database for Global Atmospheric Research. Sci Data 7, 121, 2020.

- FAOSTAT Online Statistical Service (Food and Agriculture Organization; FAO): http://faostat3.fao.org, access: October 1, 2020.

- Pandey, S., Gautam, R., Houweling, S., Van Der Gon, H. D., Sadavarte, P., Borsdorff, T., Hasekamp, O., Landgraf, J., Tol, P., Van Kempen, T., Hoogeveen, R.,

Van Hees, R., Hamburg, S. P., Maasakkers, J. D. and Aben, I.: Satellite observations reveal extreme methane leakage from a natural gas well blowout, Proc. Natl. Acad. Sci. U. S. A., 116(52), 26376–26381, doi:10.1073/pnas.1908712116, 2019.

• Schneising, O., Buchwitz, M., Reuter, M., Vanselow, S., Bovensmann, H., and Burrows, J. P.: Remote sensing of methane leakage from natural gas and petroleum systems revisited, Atmos. Chem. Phys., 20, 9169–9182, https://doi.org/10.5194/acp-20-9169-2020, 2020.

• Varon, D. J., McKeever, J., Jervis, D., Maasakkers, J. D., Pandey, S., Houweling, S., Aben, I., Scarpelli, T. and Jacob, D. J.: Satellite Discovery of Anomalously Large Methane Point Sources From Oil/Gas Production, Geophys. Res. Lett., 46(22), 13507–13516, doi:10.1029/2019GL083798, 2019.

• Zhang, Y., Gautam, R., Pandey, S., Omara, M., Maasakkers, J. D., Sadavarte, P., Lyon, D., Nesser, H., Sulprizio, M. P., Varon, D. J., Zhang, R., Houweling, S., Zavala-araiza, D., Alvarez, R. A., Lorente, A., Hamburg, S. P., Aben, I. and Jacob, D. J.: Quantifying methane emissions from the largest oil-producing basin in the United States from space, Sci. Adv., 2007(April), 1–10, 2020.

Please also note the supplement to this comment:
https://bg.copernicus.org/preprints/bg-2020-251/bg-2020-251-AC1-supplement.pdf
* * *

---

## Author Comment (AC2) · 7 Oct 2020

We thank the anonymous referee for his time and useful feedback, which helped further improve the paper. Our point-wise responses to the referee comments (in *italics*) are as follows:

**Referee**: *1. Methane emissions from ruminants in South Sudan The authors use an emission estimate from EDGAR V4.3.2 of 0.36 Tg yr-1. Firstly, there is no real discussion of uncertainties of this number, although studies in other countries have shown that significant differences between top-down and bottomup based estimates exist, even in countries with long-standing dedicated infrastructure for tracking livestock head counts, which might not be the case for a more recently formed country, such as South*

*Sudan. For example, Miller et al. 2013 (https:// www.pnas.org/ content/ 110/ 50/ 20018) have claimed that agricultural emissions are severely underestimated in the US. As the proposed domain nearly covers all of South Sudan we can try a back of the envelope calculation on what national ruminant emissions could be. According to an FAO webpage (not the ideal source, http://www.fao.org/emergencies/fao-in-action/stories/stories-detail/en/c/326186/) there are 12M cattle, 20M sheep and 25M goats in South Sudan Assuming: 5 kgCH4 yr1 for goatssheep and 35kCH4 yr-1 for cattle in developing countries this yields ca. 0.65Tg CH4, so nearly twice the EDGAR V4.3.2 estimate. Emission factors from www.tandfonline.com/ doi/ pdf/ 10.3402/ tellusb. v38i3-4.15135 It seems logical to address this issue in more detail in section 3.2.*

**Authors**: We will add a paragraph discussing the role of ruminant emissions and its uncertainty using additional sources (EDGARv5, FAOSTATS) to the manuscript. Please see our response to the first comment of the first referee.

**Referee**:  *2. Discussion of impacts of seasonally changing spatial coverage and/or clear sky bias of observations. The authors should further expand on the issue of data coverage and (potential?) clearsky bias in the manuscript. Is this a significant source of uncertainty and how was this accounted for. The seasonal decrease in emissions in JJA coincides with a significant drop in data coverage. Emissions in JJA-2019 are reported to be miniscule (statistically close to 0), but this is not appropriately addressed in the manuscript. What is happening here? Assuming that agricultural and other methane sources are still active in JJA-2019 this result is even more extraordinary.*

**Authors**: There is a correlation between the coverage and the emission estimates, however, even in JJA, there is more than 40 % of SSWR data coverage (percentage of pixels in an average map with at least 5 days of TROPOMI observations). We believe such an amount of observations should be sufficiently sensitive to the emissions from the region, unless the observations are missing over the high emission wetland's region, which is not the case. We will add the following text to clarify:

"The lowest enhancement is observed in JJA in both 2018 (10.5 ± 4.1 ppb) and 2019 (2.6 ± 3.7 ppb). It is unlikely that these low enhancements are an artefacts of the low coverage as there is still sufficient TROPOMI data (more than 40 ) and the measurement are not systematically missing over the large emission areas of SSWR, the Sudd and Machar wetlands."

In response to the second comments of the first referee, we have updated our emission quantification method to account for latitudinal XCH4 gradient and ITCZ. These updates to our method result in 1.4 ± 2.1 Tg yr$^{-1}$ total methane emissions for the JJA-2019 and 2.6 ± 3.7 ppb XCH4 enhancement. We will add the following text to our manuscript:

"We find very low total emissions in JJA-2019 (1.4 ± 2.1 Tg yr$^{-1}$), but it accommodates the season's anthropogenic emissions of about 0.48 Tg yr$^{-1}$ (sum of 2012 EDGAR emissions, and 2016 oil and gas emissions from Scarpelli et al., 2020) and GFED biomass burning emissions (0.003 Tg yr$^{-1}$)."

**Specific comments:**

**Referee**:  *Line 54: Please clarify: 38-56 higher than which RCP? They differ quite significantly in their anthropogenic emissions.*

**Authors**: It is under the strong climate mitigation scenario (RCP 2.6). We will add this information in the revised manuscript.

**Referee**:  *Line 105: Why do you choose to use Wetcharts data for 2009-2010 as basis for comparison here, although previous work you referenced (line 70) has already shown a strong trend in CH4 emissions in the Sudd wetlands after 2010, due to IE expansion?*

**Authors**: The two Wetcharts versions provide emissions for different years: Full Ensemble provides emissions for 2009-2010 and Extended Ensemble provides emissions for 2000-2015. To compare annual emissions, we use both versions of Wetcharts (Table 1). The strong trend in CH4 emissions reported by Lunt et al. (2010) is not shown by the 18-member Wetcharts Extended Ensemble, likely because it does not have the corresponding trend in the inundation extent that is derived using the ERA-interim precipitation (see Figure 4). The Full Ensemble is used for temperature dependence analysis as it provides a wider range of possible emissions (324-member ensemble).

**Referee**: *L109: The wetland extent datasets (Lehner and Doell 2004, and Bontemps et al. 2011) seem not to be very up to date and able to include any trends happening after 2010.*

**Authors**: These datasets are used by Wetcharts to calculate a baseline inundation spatial distribution, which is then scaled with ERA-interim precipitation and SWAMPS data available for years after 2010 to derive inundation extent in later years.

**Referee**: *L170: What is the temporal resolution of the meteo data set used here? Are the wind speeds from these 4 consecutive hours really independent or is the model only constrained at lower frequency intervals? This could lead to an artificially low variability. Are there any surface observations in the (wider) region to compare with the model?*

**Authors**: ERA5 meteo data has an approximately 30 km spatial resolution and hourly temporal resolution. We expect the 4 consecutive hours winds not to be independent, however, they are only used to estimate the uncertainty of the wind speed using the spread during these hours. Generally, larger hour-to-hour variation would mean a larger uncertainty in the wind speed estimate by a meteorological model for a particular hour. Surface observations are not suited for this analysis as we are interested in wind speeds over a large source area ($4 \times 10^5$ km$^2$) averaged across the whole boundary layer.

**Referee**: *Line 233: Why are emissions now 7+/-3.2 Tg yr-1 and not 7.2+/-3.2 Tg yr-1 ?*

**Authors**: That is a typo. We will correct it.

**Referee**: *Line 239: Why is the TROPOMI-based wetland CH4 emission estimate compared to a minor source such as oil and gas emissions (0.05 Tg yr-1) here? The biggest anthropogenic source in the region, even according to EDGAR, is agriculture at 0.36 Tg yr-1, which is an order of magnitude bigger than OG (possibly bigger, see general comments).*

**Authors**: Oil and gas emissions can be quite uncertain. This is especially true in regions with less strict flaring regulation. In addition, they are located in the SSWR and may correlate spatially with wetland emissions. The agricultural methane emissions in SSWR due to agriculture waste burning and agriculture soils are negligible as per EDGAR. To address emissions from enteric fermentation (0.36 Tg/yr), we will add new text to the revised manuscript. Please refer to our response the first comment of the first referee.

**Referee**: *Line 246: Please correct to 'nearly an order of magnitude lower'.*

**Authors**: Done

**Referee**: *Line 252: It seems the upper end of the WetCharts ensemble is not that far off the lower end of your estimate here, so claiming they differ by an order of magnitude seems unhelpful.*

**Authors**: In general, the mean emission estimates of the bottom-up models are an order of magnitude lower. We will clarify this in the revised manuscript.

"SSWR integrated mean methane emission estimates from the process models are nearly an order of magnitude lower than those from TROPOMI"

**Referee**: *Line 287: JJA-2019 seems extremely low, this needs to be discussed further (see general comments).*

**Authors**: Please see our response to the second comment of the referee.

**Referee**: *Line 297: It is unclear if TROPOMI-derived CH4 emission estimates really track river water heights all that well or if the effect is mostly Q10 related. Maybe those two components should not be singled out or it should be made clear early in the manuscript that both components contribute.*

**Authors**: We address both the issues as a possible explanation one by one in the manuscript without claiming that one of them is the definite cause. To clarify that both the factors contribute, we will add the following sentence to the manuscript:

"The seasonality difference of models and TROPOMI emissions can be explained by a combination of temperature and inundation extent information used in the models."

**Referee**: *Temperatures are surely lower in JJA, but not low enough to explain the emissions reported for JJA-2019.*

**Authors**: The relation between temperature and emissions is not linear. Therefore, a small change in temperature can cause a large emission change.

**Referee**: *Line 362: Some models report emissions that are definitely NOT 'an order of magnitude' smaller.*

**Authors**: The mean estimates of the models are roughly an order of magnitude smaller. We will clarify it.

**Referee**: *Line 365: One model ensemble member with Q10 = 3 estimates 3.7 Tg yr-1 despite the 'poor IE estimates'. So it seems this study cannot disentangle the two issues, which should be reflected in the conclusion here. Furthermore, after reading this manuscript it seems more convincing that Q10 is the key issue rather than IE.*

**Authors**: The mean of Wetcharts Full Ensemble emission estimates with Q10 = 3 is 1 Tg yr$^{-1}$ which is still nearly an order of magnitude lowered than our emissions estimates. We do not claim to disentangle the influence of inundation extent and temperature. We only try to hypothesize possible causes that explain the difference between TROPOMI and model emission phasing based on shortcomings of assumptions and
input dataset used by the models.

**Referee**: *Table 1: Please elaborate what the variable 'data coverage' signifies here. For example, for DJF-2018: is it 91 of all cells were covered at least once in this 3 month period or were 91 of all possible data collected for an average cell. For example, if you measure every cell at least once in DJF-2018 would you label this as 100 coverage (which it is not) or do you need to measure all cells, all of the time (once per day) to reach 100 coverage?*

**Authors**: We agree with the referee that data coverage needs a proper description. We will add the following text to clarify:

"Data coverage is defined here as the fraction of $0.1 \times 0.1$ degree grid cells in SSWR with at least five days of high-quality TROPOMI measurement in a quarter."

---

## Author Response (AR1)

Dear editors,

Please find attached the revised manuscript. The change made to the manuscript are shown in this document. To address your comment on emissions due to livestock and manure management, also raised by the first reviewer, we have added the following text to the revised manuscript:

Manure management:

[revised manuscript text omitted]

---

## Author Response (AR2)

Dear editors,

Please find attached the revised manuscript. The changes made to the manuscript are shown in this document. To address your comment on emissions from wild animals and domestic ruminants, we have added the following text to the revised manuscript:

[revised manuscript text omitted]